# Characterization of *Euglena gracilis* Mutants Generated by Long-Term Serial Treatment with a Low Concentration of Ethyl Methanesulfonate

**DOI:** 10.3390/microorganisms13020370

**Published:** 2025-02-08

**Authors:** Ji-Yeon Kang, Younglan Ban, Eui-Cheol Shin, Jong-Hee Kwon

**Affiliations:** 1Division of Applied Life Sciences (BK21), Gyeongsang National University, Jinju 52828, Republic of Korea; kangz0627@naver.com; 2Department of Food Science & Technology and Institute of Agriculture & Life Science, Gyeongsang National University, Jinju 52828, Republic of Korea; 3Department of GreenBio Science/Food Science and Technology, Gyeongsang National University, Jinju 52828, Republic of Korea; byr1218@naver.com (Y.B.); eshin@gnu.ac.kr (E.-C.S.)

**Keywords:** *Euglena gracilis*, EMS mutagenesis, mutant selection, paramylon, volatile compounds

## Abstract

*Euglena gracilis* is a microalga that has great promise for the production of biofuels, functional foods, and bioactive compounds, and mutagenesis and effective screening methods are required to develop *Euglena* strains that have industrial use. Ethyl methanesulfonate (EMS) is a widely used mutagen, but is highly lethal to *Euglena* at typical concentrations. In the present study, low-concentration, long-time EMS exposure combined with serial treatment was introduced for generating *Euglena* mutants. We then used screening protocols to select cells with altered motility or pigmentation, and isolated two distinct strains of *Euglena*: Mutant 333 and Mutant 335. Mutant 333 showed increased motility but exhibited a decreased differentiation rate and reduced paramylon content (13.5%), making it unsuitable for industrial applications. However, Mutant 335, which had a deficiency of chlorophyll, had a high paramylon content (31.62%) and a mild and pleasant odor profile due to decreased concentrations of certain volatile compounds, with confirmation by GC-MS analysis. The Mutant 335 strain is suitable for the production of functional food products and renewable jet fuel.

## 1. Introduction

*Euglena gracilis* is a green microalga known as a potential candidate for the production of commercial products that can be used in the field of food, fiber, feed, fertilizer, and fuel [1]. As an example in the food industry, these cells can produce many high-value metabolic products, including vitamins, amino acids, pigments, unsaturated fatty acids, and carbohydrates [1,2,3]. Especially, *Euglena gracilis* can produce unique substances, such as paramylon and wax esters, that cannot be biosynthesized by other microorganisms. Paramylon is a linear β-1,3-glucan that has potential as a functional product due to its immune-enhancing, cholesterol-reducing, and antioxidant activity [4,5,6,7,8]. *Euglena* also converts paramylon into wax esters in the cytoplasm under anaerobic conditions, and these wax esters are composed entirely of saturated carbon chains, including myristyl myristate (C14:0–C14:0), which is an ideal raw material for a drop-in bio jet fuel [9,10]. In addition, *Euglena gracilis* possesses the ability to survive in extreme conditions, such as low pH environments, which facilitates its dominance in cultivation systems [11].

Various mutagenesis techniques and screening methods can be used to develop strains of *Euglena* with increased biomass productivity, harvesting efficiency, or metabolic flux, and these strains may be suitable for various industrial uses [4,12]. Targeted genetic engineering and random mutation methods can be used to alter the traits of cells [8]. For example, a recent study used the CRISPR-Cas9 method to create a non-motile mutant of *Euglena* that enabled increased harvest efficiency due to natural precipitation [13]. On the other hand, random mutagenesis has been widely used to generate microalgae with different properties. Random mutagenesis, which can be achieved using ultraviolet (UV) radiation or exposure to ethyl methanesulfonate (EMS), has the advantages of being simple, fast, and inexpensive [12,14]. Previous studies have induced random mutagenesis in *Euglena* using Fe ions, UV radiation, and heavy particles and then screened for mutants with improved growth rate, increased lipid content, and increased paramylon content [4,13,15,16]. The chemical mutagen EMS is an alkylating agent, which reacts with purine bases, particularly adenine (A) and guanine (G), leading to incorrect base pairing during DNA replication [17]. In principle, EMS can easily affect DNA in *Euglena*. However, very few previous studies have used EMS mutagenesis to develop useful strains of *Euglena*.

In this study, we introduced a series of EMS treatment processes in *Euglena gracilis* and then screened for mutants that exhibited differences in phenotypic characteristics. This procedure led to the identification of one mutant that was completely deficient in chlorophyll and another mutant that had greatly increased motility. We then characterized the growth rate, paramylon content, and odor analysis.

## 2. Material and Methods

### 2.1. Cultivation of Cells

*Euglena gracilis* UTEX367 (henceforth wild type, WT) was obtained from the UTEX Culture Collection of Algae (University of Texas, Austin, TX, USA) and was cultivated in modified YM liquid medium with cool white fluorescent bulbs at a photon flux density of 50 μmol photons m^−2^ s^−1^ (photosynthetically active radiation) at room temperature with agitation at 100 rpm. The modified YM liquid medium consisted of the commercial YM broth (yeast extract 1.5 g L^−1^, malt extract 1.5 g L^−1^, peptone 2.5 g L^−1^, dextrose 5 g L^−1^) and 1 mL L^−1^ mineral stock (50 mg L^−1^ Na_2_EDTA∙2H_2_O, 50 mg L^−1^ ZnSO_4_∙7H_2_O, 5 mg L^−1^ MnCl_2_∙4H_2_O, 5 mg L^−1^ FeSO_4_∙7H_2_O, 2 mg L^−1^ CuSO_4_∙5H_2_O, and 1 mg L^−1^ [NH_4_]_6_Mo_7_O_24_∙4H_2_O).

### 2.2. EMS Treatment, Determination of Cell Survival, and Screening for Mutants

For EMS mutagenesis, *Euglena* cells in the early-exponential growth phase (3.17 × 10^5^ cells mL^−1^) in a 50 mL T-flask were exposed to 0.008, 0.012, 0.016, 0.024, 0.032, and 0.040 M EMS (Sigma-Aldrich), and then observed under a microscope at 24 h intervals for 5 days. The percentage of motile of cells was scored from “−” (0%) to “+++++” (100%), based on these daily observations. The survival rate of cells was determined by spreading a 10 μL sample onto an YM agar plate, using the same light and temperature conditions as above. After 2 weeks, the number of colonies was calculated, and cell viability was expressed relative to a control group (no EMS). A concentration of EMS that yielded a survival rate below 5% was considered suitable for mutagenesis and screening [18].

Then, colonies that were lighter green than the WT were selected, transferred into a 50 mL T-flask, and cultured in a modified YM liquid medium. During the screening process, the standard YM agar and a modified YM agar (which contained half the amount of agar powder) were prepared to characterize cell movement. Colonies that were colorless were screened on the modified YM agar, transferred into a 50 mL T-flask, and cultured in the modified YM liquid medium for determination of the growth rate.

Under the selected conditions, a living colony from the first EMS treatment that had no phenotypic differences from the WT was chosen and re-cultivated for a second EMS treatment. This serial EMS treatment was repeated until a colony exhibited the desired trait: decreased pigmentation (subsequently named *Euglena* Mutant 335) and increased motility (subsequently named *Euglena* Mutant 333).

### 2.3. Characterization of Euglena Mutants

To evaluate characteristics related to cell mobility and heterotrophic cell division, a 10 µL drop of the two types of mutant cells and WT cells (each with the same cell count), was added to a modified YM agar plate. To measure the cell motility, the average diameter of each colony was measured after 24 h. To measure the rate of cell division, all cells derived from one spot were collected using 5 mL of distilled water after 7 days, and the total number of cells was counted using a hemocytometer. The experiments were performed in triplicate. To measure the cell growth rate in a liquid culture, the screened mutants were cultivated in a modified YM liquid medium in 50 mL T-flasks under continuous illumination as above. Cell growth was then monitored by measuring the optical density at 680 nm (OD_680nm_).

To evaluate photosynthetic growth, the two mutants and WT cells were cultivated in culture flasks with a glucose-free medium containing 0.5 g L^−1^ NH_4_Cl, 0.1 g L^−1^ MgSO_4_∙7H_2_O; 0.1 g L^−1^ CaCl_2_∙2H_2_O; 1 mL L^−1^ phosphate stock consisting of 0.1 g L^−1^ NaH_2_PO_4_ and 50 mg L^−1^ K_2_HPO_4_; and 1 mL L^−1^ of a mineral stock (described above). Continuous illumination was provided from a flat LED panel as above. All growth media were sterilized by autoclaving, with maintenance of a temperature of 121 °C for at least 20 min.

### 2.4. Measurement of Chlorophylls

The chlorophyll a (Chl a) and chlorophyll b (Chl b) levels of cells were determined using a dimethyl sulfoxide (DMSO) extraction method [19]. First, 1 mL of cells was centrifuged at 13,000 rpm for 5 min, and the pellet was suspended in 1 mL of DMSO and incubated at 60 °C for 40 min. The samples were then centrifuged at 13,000 rpm for 5 min, and the optical density of the supernatant was measured at three wavelengths (OD_480nm_, OD_649nm_, OD_665nm_) using a UV–Vis spectrophotometer (JASCO V-730, Tokyo, Japan) [18]. The concentrations were then determined using the following two equations:Chl a [μg mL^−1^] = 12.19 × OD_665_−3.45 × OD_665_(1)Chl b [μg mL^−1^] = 21.99 × OD_665_−3.45 × OD_649_(2)

### 2.5. Measurement of Paramylon Using Gravimetry

*Euglena* cells were grown in a modified YM medium. First, 50 mg of a freeze-dried sample was weighed and suspended in 7 mL of acetone to remove chlorophyll, and the cells were then lysed by sonication for 10 min. This acetone and sonication process was then repeated, the cells were centrifuged at 4000 rpm for 5 min, the sample was suspended in 1% sodium dodecyl sulfate (SDS) solution to remove substances other than paramylon, and the sample was then boiled at 100 °C for 30 min. The sample was then cooled to room temperature, boiled again in SDS, and washed twice using distilled water. Finally, the sample was transferred to a 1.5 mL microtube and centrifuged at 4000 rpm for 5 min. The supernatant was discarded, the sample was dried overnight at 80 °C, and the weight was measured. The paramylon content (%) was calculated by dividing the paramylon weight by the dry cell weight [4].

### 2.6. Measurement of Paramylon Using Aniline Blue Staining

*Euglena* cells were grown in a modified YM medium. First, 20 mg of a paramylon standard was dissolved in 40 mL of 1M NaOH to make a standard solution. Then, the standard was heated at 80 °C for 5 min and mixed with 0.2 M glycine in a ratio of 1:7. Next, 120 µL of a standard containing 0.2 M glycine and 80 µL of 0.1% aniline blue was added to a black microplate. The microplate was heated at 80 °C for 15 min, cooled for 15 min (at room temperature in a dark room), and the fluorescence was determined using a microplate reader (Infinite 200 Pro, TECAN, Vienna, Austria), with excitation at 400 nm and an emission cutoff at 505 nm.

For analysis of the samples, lyophilized cells were placed in a conical tube (10 mg), 1 M NaOH was added (10 mL), and the sample was then heated at 80 °C for 5 min and vortexed. Then, 1 mL of the heated sample was mixed with 7 mL of a 0.2 M glycine solution and vortexed. Next, 120 µL of the sample was mixed with glycine and added to a 96-well black microplate. Then, 80 µL of 0.1% aniline blue was added to each well and the sample was maintained in darkness. The sample was then heated at 80 °C for 15 min, cooled for 15 min at room temperature, in a dark room, and the fluorescence was determined as described above [4].

### 2.7. Analysis of Volatile Components

The volatile components of each sample were analyzed using GC-MS (Agilent 7890A & 5975C, Agilent Technologies, Palo Alto, CA, USA). These experiments were performed using the headspace analysis method and 50/30 µm, divinylbenzene/carboxen/polydimethylsiloxane (DVB/CAR/PDMS)-coated solid-phase microextraction (SPME) fibers (Supelco Inc., Bellefonte, PA, USA). A 2 g sample was placed in a vial, sealed with an aluminum cap, and equilibrated at 60 °C for 20 min. The SPME fiber was then exposed to the inside of the vial for 25 min. The SPME fiber was adsorbed and analyzed using a system equipped with an HP-5MS column (30 m × 0.25 mm, id: 0.25 um film thickness, Supelco, Inc.). The injector temperature was 220 °C, the flow rate was 1.0 mL min^−1^, the carrier gas was helium, and the analysis was performed under splitless analysis conditions. The oven temperature was maintained at 40 °C for 5 min, then increased to 200 °C at a rate of 5 °C min^−1^. Volatile compounds isolated from the total ion chromatogram (TIC) were identified by reference to a mass spectrum library. For quantitative analysis, the peak area of each volatile component was determined and converted to µg/kg after comparison with the internal standard (C15:0, pentadecane). The separated volatile substances were also analyzed using an olfactory detection port (ODP-Ⅲ, Gerstel, Inc., Linthicum, MD, USA) that was equipped on the GC-MS. The intensity of the perceived odor was scored from 1 to 4, with a higher number indicating a stronger odor. The intensity and duration of the fragrance were also recorded.

## 3. Results and Discussion

### 3.1. EMS Mutagenesis

EMS is widely used in the development of microalgae strains due to its advantages in simple and efficient mutant generation. There is increasing attention devoted to microalgae an eco-friendly feedstock for biodiesel production because their photosynthesis can increase biomass and lead to the accumulation of abundant lipids. For example, EMS mutations in *Chlorella vulgaris* resulted in enhanced photosynthetic efficiency under high light conditions (200 μmol photons m^2^ s^−1^) and a 44.5% increase in biomass production [18]. In another study, the application of EMS mutation to *Botryococcus braunii* AARL G036 enhanced biomass, lipid, and hydrocarbon productivity by maximizing lipid (up to 48.6%) and hydrocarbon (up to 71.6%) accumulation levels in the biomass [20]. Meanwhile, various studies are actively being conducted to develop oxygen-resistant hydrogenase for biohydrogen production [21].

However, very few studies have used EMS mutagenesis with *Euglena*. The typical concentration of EMS used for the mutagenesis of microalgae is 0.5 to 3 M, and EMS is then typically washed off after 1 to 4 h [22,23,24,25]. But our preliminary application of this standard method with 0.5 to 3 M EMS completely killed all *Euglena* cells. Therefore, we aimed to develop a method for transforming and screening *Euglena* by treating it with EMS at a 10-fold lower concentration for 72 h, which resulted in a survival rate to below 5%. Long-term treatment with EMS is a critical strategy for increasing the frequency of mutation induction, allowing for the acquisition of a diverse range of mutants [26]. Although prolonged EMS exposure enhances the probability of inducing specific mutations, excessive exposure may compromise cell survival. Therefore, it is crucial to balance concentration and duration to maximize mutation efficiency depending on the species. In addition, we applied a serial EMS treatment, in which the surviving colonies were re-cultured after one EMS treatment, and then re-treated with the same level of EMS until the desired phenotype (altered motility or pigment content) was observed. This method increased the number of mutant cells that could be used for experiments.

Thus, we first treated *Euglena* cells with different concentrations of EMS (0, 0.008, 0.012, or 0.016 M) and then observed cell motility under a microscope for the following 5 days (Table 1). A 2 h treatment at all tested concentrations had no effect on motility. However, beginning at 24 h, the cells had decreased motility according to the concentration of EMS and time of treatment. After 48 hours, *Euglena* activity (expressed as its motility) was not observed in the 0.016 M EMS condition, and almost all cells were rounded and coiled. In the condition treated with 0.012 M EMS, activity began to decrease rapidly after 48 hours, and no activity was observed at this concentration after 96 hours. Cells treated with 0.008 M EMS for 96 h had slightly lower activity than the control, but more than 20% of these cells were still actively moving.

In addition to monitoring changes in cell activity, we measured cell survival by determining CFUs (Figure 1). As a result, treatment with 0.008 EMS led to a survival rate of 85.19% at 2 h, but the survival rate was 11.11% on the fourth day. Treatment with 0.016 M EMS led to complete cell mortality on day 2, and treatment with 0.012 M EMS led to a survival rate of 3.33% on day 3. Other studies that examined EMS mutagenesis of microalgae reported survival rates from 0.142% to 26% [23,24,25]. In this study, *Euglena* showed a survival rate of 3.33% when treated with 0.012 M EMS for 72 hours (Figure 1). Interestingly, the reduction in *Euglena* motility (Table 1) and survival rate (Figure 1) shows a similar pattern, proportional to the EMS treatment concentration and duration. After treatment with 0.016 M EMS for 48 hours, *Euglena* motility was not observed under the microscope (Table 1), and no *Euglena* colonies were detected (Figure 1). When *Euglena* exhibited ++ (about 40%) motility, its survival rate ranged from 15.60% to 18.15%. Therefore, it is concluded that the optimal conditions required for mutation induction using EMS can be sufficiently estimated through comparative observation of *Euglena* activity via microscopy, even without verifying survival rates through a CFU test.

### 3.2. Screening EMS Mutants

Colonies that survived the 72 h treatment with 0.012 M EMS were isolated, re-cultured, and subjected to identical EMS treatments until the resulting strains exhibited distinct changes in color, morphology, or motility. The sequence of three EMS treatments ultimately led to the isolation of one strain with increased motility (Mutant 333) and another strain with chlorophyll deficiency (Mutant 335).

To select strains with increased motility, we used a YM agar plate that only contained 7 g/L of agar powder, leading to a soft solid agar region that did not restrict cell movement. To compare the motility of mutant strains and the WT, we dropped 10 μL of the same number of mutant and WT *Euglena* cells (OD_680nm_ = 0.5) at the center of an agar plate and observed the cells every 2 days (Figure 2). After 3 days, the colony of Mutant 333 was much larger than that of WT, and after 7 days Mutant 333 had a colony diameter of 7.0 cm, nearly 1.3 times larger than that of the WT (5.2 cm).

To investigate the relationship between mobility and cell division, the two strains (WT and Mutant 333) were grown on plates for 7 days, collected, and the cell count was measured under a microscope. The WT cells had a concentration of 1.76 × 10^5^ ± 47 cells mL^−1^ and the Mutant 333 cells had a concentration of 0.74 × 10^5^ ± 205 cells mL^−1^. Thus, Mutant 333 had increased motility, but decreased production of biomass. Because Mutant 333 was generated through random mutagenesis, its unique phenotype (increased motility and decreased growth) could be attributed to multiple mutations. We also isolated Mutant 335, which had a notable chlorophyll deficiency, after three successive EMS treatments (Figure 3). This mutant had the lowest motility and rate of cell division.

Liquid culture is commonly used for the large-scale production of valuable compounds in *Euglena*. If the results of liquid culture can be inferred from the growth rate on agar plates, it would save significant time and effort in the selection of strains capable of rapidly accumulating biomass in a liquid medium among the many mutants generated under EMS treatment conditions. Thus, we used modified YM liquid medium to grow WT, Mutant 333, and Mutant 335 cells under identical conditions (Figure 4). The WT cells began to grow rapidly after about 10 days and reached a maximum absorbance at 20 days (OD_680nm_ = 4.3). In contrast, Mutant 333 reached only about 50% of the WT value after 30 days. Mutant 335 exhibited a lower accumulation of biomass, presumably related to chlorophyll deficiency. Our measurements also demonstrated that the chlorophyll content was slightly greater in Mutant 333 than in WT, but that Mutant 335 had extremely low or undetectable levels of chlorophyll (Figure 5).

Despite the slightly greater chlorophyll content of Mutant 333, it had the lowest growth rate under mixotrophic conditions. However, under phototrophic conditions in a static culture (no mixing), Mutant 333 had a slightly better growth rate than WT during the later stages of cultivation (Figure 6). In particular, WT had greater growth during the initial 16 days, but Mutant 333 had greater growth after 16 days. According to the Beer–Lambert law, when the absorbance is greater than 0.2, the initial light intensity decreases by 50% in the center [27]. Therefore, in various experiments optimizing cultivation conditions through continuous culture at a constant concentration, the cell density value is adjusted to not exceed an OD_680nm_ value of 0.2 [28,29]. In the cultivation of *Euglena*, without physical agitation and with light provided from above, the mixing of internal cells during cultivation relies solely on the motility of *Euglena*. Based on the results shown in Figure 6, when the OD_680nm_ is below about 0.2, the shading caused by cells within the cultivation space is minimal. Therefore, regardless of *Euglena'*s motility, all *Euglena* cells in the space can evenly utilize the available light without the issues. As a result, the wild-type (WT), which has a faster division rate than Mutant 333, exhibits a steep growth rate. When the OD_680nm_ value exceeds 0.3, the increase in dark zones caused by self-shading results in light utilization being dependent on the motility of *Euglena*. Consequently, although the WT has a higher division rate, the reduced light utilization allows Mutant 333 to achieve a higher growth rate. As an example of a study demonstrating how the efficiency of *Euglena* growth changes depend on light utilization, the growth rate and productivity of *Euglena gracilis* improved as the proportion of light exposure increased; the highest productivity was observed under the 16:8 light/dark cycle condition compared to conditions with shorter light cycles [30]. Therefore, we attribute these results to the greater motility of Mutant 333 under phototrophic conditions without mixing. Moreover, these results demonstrate that increased cell motility on solid media corresponds to increased motility in liquid media. Our examination of Mutant 335 indicated no growth under phototrophic conditions in a static culture.

### 3.3. Paramylon Production

We used two methods to measure the accumulation of paramylon in *Euglena* cells: gravimetry and aniline blue staining (Figure 7) [4]. The paramylon content of WT was 20% using gravimetry and 18% using aniline blue staining. The paramylon content of Mutant 333 was approximately 15% using gravimetry and 12% using aniline blue staining, lower values than WT. Notably, Mutant 335 had about 1.72-fold more paramylon than WT: approximately 30% by gravimetry and 35% by aniline blue staining. The lower paramylon content in Mutant 333 would be likely a consequence of its enhanced motility. A previous study described a *Euglena* mutant with flagella defects generated by Fe-ion irradiation that accumulated approximately 1.6 times more paramylon than its WT. This study explains the increase in paramylon accumulation as being caused by the energy conservation by not swimming [31]. As in another study, in *Chlamydomonas reinhardtii* flagellar-deficient mutants, starch accumulation increased due to the absence of energy-intensive processes like flagellar regeneration [32]. Through reverse results caused by a completely opposite factor, the greater energy consumption used for motility may have reduced the accumulation of paramylon, a carbohydrate storage compound. However, the possibility of changes in the paramylon content caused by other genetic defects resulting from EMS treatment cannot be ignored, and further research is needed.

In the present study, the pigment-deficient Mutant 335 accumulated approximately 1.72 times more paramylon than WT. The deficiency in pigments such as chlorophyll reduces the efficiency of photosynthesis, potentially redirecting the carbon flux towards storage compounds like paramylon as a compensatory mechanism [18]. Pigment-deficient mutant exhibits increased carbon storage as paramylon, likely as an adaptive response to reduced light energy capture. Under pigment-deficient conditions, cells might prioritize the accumulation of paramylon to cope with metabolic stress and maintain cellular homeostasis [29]. In other words, the reduced pigment content may minimize the energy demand required for light harvesting, resulting in the promotion of the synthesis and accumulation of storage carbohydrates such as paramylon. Another study examined a naturally occurring mutant of *Euglena gracilis* (WZSL) that lacked chloroplasts and cannot perform photosynthesis. Under dark conditions and using glucose as a carbon source, this mutant accumulated approximately three times more paramylon than the WT [33]. A similar study conducted under the same conditions reported the WZSL strain accumulated approximately 1.8 times more paramylon than WT [34].

### 3.4. Volatile Compounds

Microorganisms that are cultivated for industrial applications often contain unique aromatic compounds, and the presence or absence of these compounds can affect the application and commercialization of microalga-derived products in the food and cosmetic industries [35]. Particularly, some microalgae can emit strong ‘fishy’ odors, which limits their commercial utilization [36]. Thus, we compared the composition and content of volatile and aromatic compounds in WT and the pigment-deficient Mutant 335.

Our initial assessment indicated that Mutant 335 had a less ‘fishy’ odor than WT. To verify this and analyze the aroma component profiles of the mutants, we conducted aroma analysis that specifically compared these two strains. We first identified all volatile compounds using GC/MS (see Appendix A). The results indicate that Mutant 335 produced fewer types of volatile compounds than WT, and that the concentrations of these compounds were generally lower. Notably, the levels of aldehydes and sulfur compounds, which are known to cause ‘fishy’ odors, are much lower in Mutant 335. In contrast, the WT contained high concentrations of many volatile compounds such as alcohols, aldehydes, sulfur compounds, and hydrocarbons, and this was likely responsible for its strong and complex ‘fishy’ aroma. Photosynthesis is not possible in pigment-deficient Mutant 335, which may be a key factor in the changes observed in the produced volatile compounds. More volatile compounds are produced in oils like linoleic acid when chlorophyll is present. In principle, chlorophyll can generate singlet oxygen from triplet oxygen upon exposure to fluorescent light, which plays important roles in the formation of 2-pentylfuran, *trans*-2-heptenal, and 1-octen-3-ol in linoleic acid [37]. 1-Octen-3-ol is one of the volatile compounds found in the WT in our *Euglena*-related research but is not detected in the pigment-deficient Mutant 335. In Mutant 335, chlorophyll-photosensitized reactions do not occur, which may explain the reduced concentration and number of volatile compounds observed compared to the WT. However, the possibility of the production of a volatile compound caused by other genetic changes resulting from EMS treatment cannot be ignored; it is also limited to explain the extensive changes in the volatile compound profile caused solely by genetic variations. In the future, identification of the responsible gene in the mutants can provide a clearer understanding of the relationship between the associated phenotypes and changes in the paramylon and volatile compound content.

Table 2 presents the results of the volatile organic compound (VOCs) analysis for Mutant 335 and WT based GC-olfactory analysis of the 127 volatile compounds listed in Appendix A. Notably, Mutant 335 had no ‘mugwort’ scent (1-octen-3-ol, 2-methyl-1-heptene, styrene), and lacked the ‘barley tea’ and ‘green tea’ scent (hexanal, hexyl formate, 4-ethylbenzoic acid) that was present in WT. However, compounds associated with a ‘herbal’ scent (2-methyl-1-undecanol) and a ‘green’ scent (2,4,6-Trimethyldecane) had higher concentrations in Mutant 335. This suggests that the decrease or modification in specific aroma compounds in Mutant 335 may make it suitable for various food products, such as tea, salad dressings, and herb-based foods.

## 4. Conclusions

This study successfully established an EMS-based mutagenesis method for *Euglena gracilis* by comparing the effects of different mutagenesis conditions. We induced mutations using three sequential treatments with EMS at a concentration of 0.012 M over 72 h, and isolated one strain with increased motility (Mutant 333) and another strain with chlorophyll deficiency (Mutant 335). In addition, the changes in cell division in the mutants were similar on agar plates and a liquid medium. This demonstrates the feasibility of isolating strains of *Euglena* that have altered motility and growth rate on plates, without the need for a liquid culture. Mutant 333 had a 1.4 times lower paramylon content than the WT, and Mutant 335 (which lacks chlorophyll) had 1.72 times more than WT. Moreover, our analysis of volatile substances showed that Mutant 335 had lower levels of aldehydes and sulfur compounds than the WT, so that it had a less ‘fishy’ smell and was therefore more suitable for applications in the food industries.

## Figures and Tables

**Figure 1 microorganisms-13-00370-f001:**
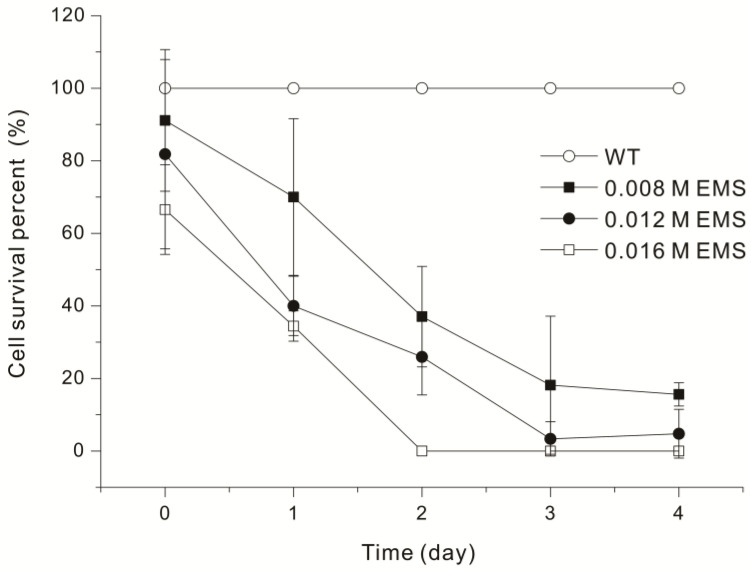
Survival of *Euglena gracilis* UTEX367 (WT) after treatment with different concentrations of EMS (0.008, 0.012, 0.016 M) for different durations. Data were analyzed using one-way ANOVA followed by Duncan’s test (*p* < 0.05).

**Figure 2 microorganisms-13-00370-f002:**
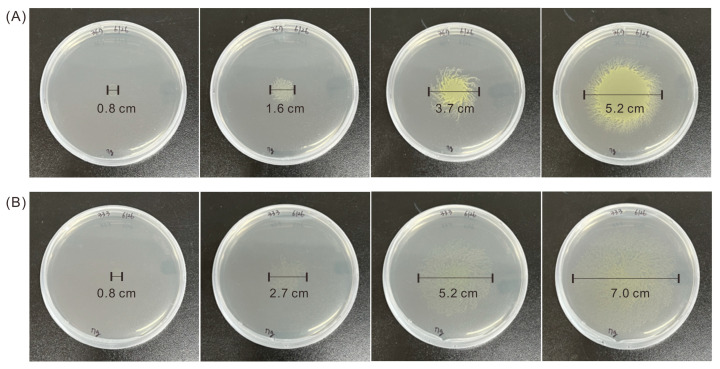
Motility of *Euglena* Mutant 333 (**A**) and WT (**B**) on agar plates after cultivation for 1, 3, 5, and 7 days (left to right).

**Figure 3 microorganisms-13-00370-f003:**
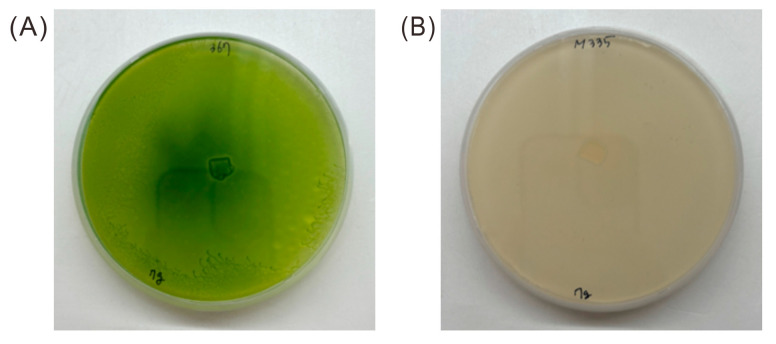
Appearance of *Euglena* WT (**A**) and Mutant 335 (**B**) after growth on agar plates for 30 days.

**Figure 4 microorganisms-13-00370-f004:**
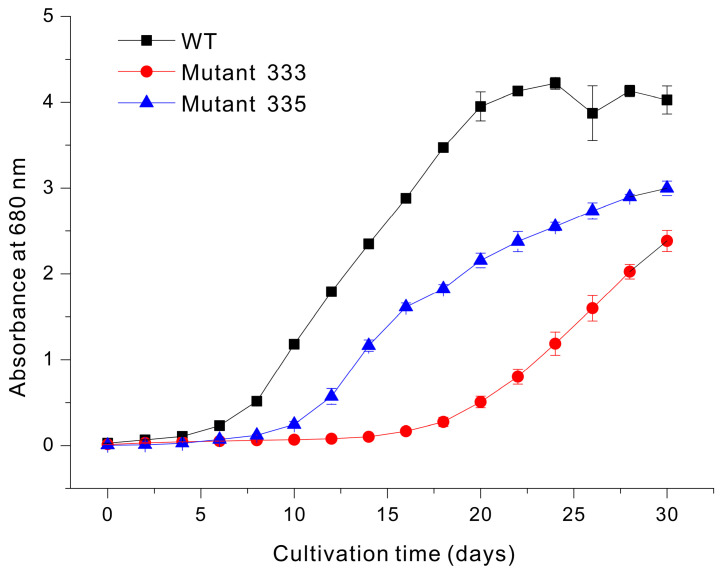
Growth of *Euglena* WT, Mutant 333, and Mutant 335 in modified YM liquid medium.

**Figure 5 microorganisms-13-00370-f005:**
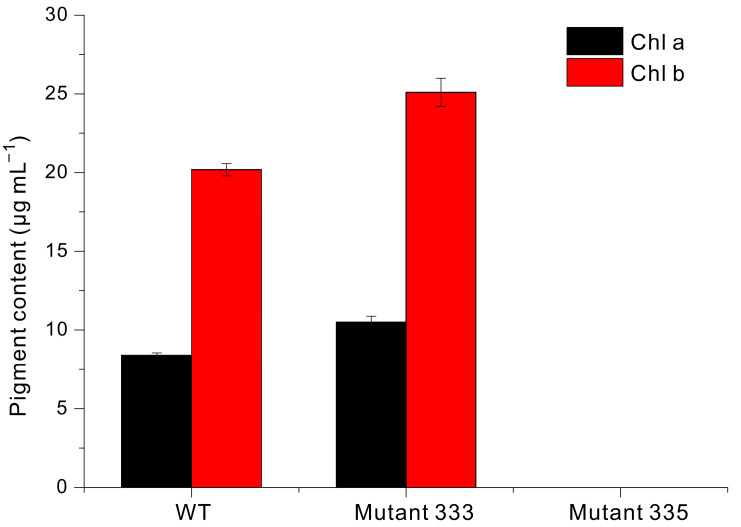
Levels of chlorophyll a and chlorophyll b in *Euglena* WT, Mutant 333, and Mutant 335.

**Figure 6 microorganisms-13-00370-f006:**
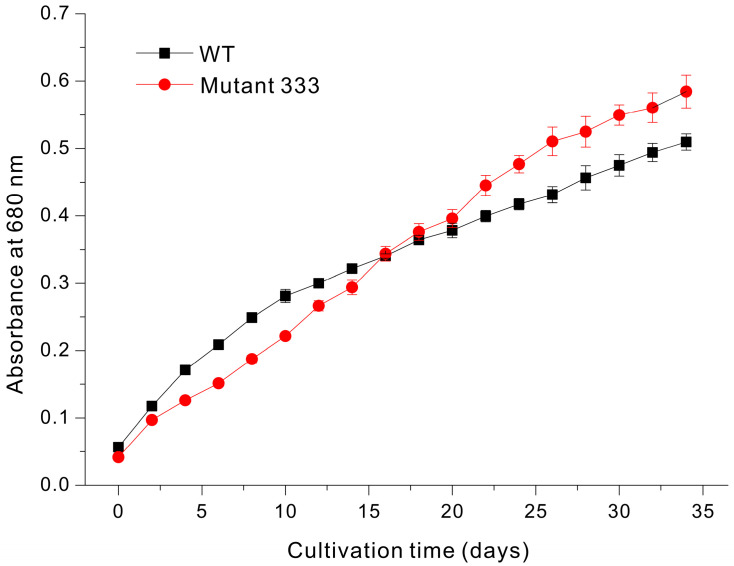
Growth of *Euglena* WT and Mutant 333 in glucose-free liquid medium.

**Figure 7 microorganisms-13-00370-f007:**
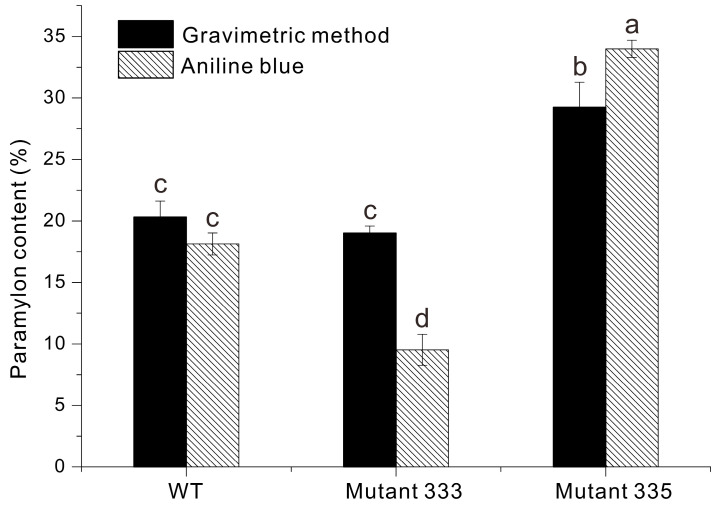
Paramylon content of *Euglena* WT, Mutant 333, and Mutant 335 determined by gravimetry and aniline blue staining. Data were analyzed using one-way ANOVA followed by Duncan’s test (*p* < 0.05). Different lowercase letters at the top of the bar graph indicate statistically different between groups (*p* < 0.05), and the same lowercase letter indicates no significant difference (*p* > 0.05). n = 3.

**Table 1 microorganisms-13-00370-t001:** Effect of EMS concentration and duration of treatment on motility of *Euglena gracilis* *.

EMS Concentration	2 h	24 h	48 h	72 h	96 h
0 M (Control)	+++++	+++++	+++++	+++++	+++++
0.008 M	+++++	+++++	++++	++	++
0.012 M	+++++	++++	++	+	−
0.016 M	+++++	+++	−	−	−

*, + indicates percentage of motile cells (+++++: 100%; ++++: 80%; +++: 60%; ++: 40%; +: 20%; −: 0%).

**Table 2 microorganisms-13-00370-t002:** Odor intensity in *Euglena* WT and *Euglena* Mutant 335 based on GC olfactometry.

Volatile Compound	RT ^(1)^	Relative Intensity		Odor Description
	(min)	Mutant 335	WT	
**Alcohols**				
1-Hexanol	10.29	0	1	Green
1-Octen-3-ol	13.83	0	2	Mugwort
2-Methyl-1-undecanol	20.23	2	0	Herbaceous
**Aldehydes**				
Hexanal	7.99	0	1	Green tea
**Acid and esters**				
Hexyl formate	10.21	0	2	Barley tea
4-Ethylbenzoic acid	11.31	0	2	Green tea
**Hydrocarbons**				
2-Methyl-1-heptene	10.34	0	1	Mugwort
Styrene	10.98	0	1	Mugwort
2,6-Dimethyloctane	12.33	0	2	Roasted
2,4,6-Trimethyldecane	21.84	2	1	Green
1,3-Bis(1,1-dimethylethyl)-benzene	22.03	1	1	Herbaceous

^(1)^ RT: retention time.

## Data Availability

The original contributions presented in this study are included in the article/Appendix A. Further inquiries can be directed to the corresponding author.

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
