# Peer review of "Characterization of Euglena gracilis Mutants Generated by Long-Term Serial Treatment with a Low Concentration of Ethyl Methanesulfonate"

_microorganisms, 2025, doi:10.3390/microorganisms13020370_

Round 1

Reviewer 1 Report

Comments and Suggestions for Authors

The manuscript investigates the development of mutant strains of Euglena gracilis using ethyl methanesulfonate (EMS) mutagenesis. While the experiments are well-executed, some revisions are necessary before publication:

  1. The importance and advantages of prolonged EMS exposure should be elaborated upon with appropriate references. The novelty of the manuscript cannot rely solely on microalgal mutation; additional context is needed to highlight its significance.
  2. Figures 1 and 2 suggest that microscopy observations are sufficient to assess mutation conditions. However, the explanation provided in the text is inadequate and requires further elaboration to support this conclusion.
  3. Statistical analysis results should be included in Figure 7 to enhance the reliability and interpretation of the data.
  4. The significance of the GC data and its implications for the commercialization potential of the mutants should be discussed in greater detail.
Comments on the Quality of English Language

 The English could be improved to more clearly express the research.

Author Response

Reviewer 1

The manuscript investigates the development of mutant strains of Euglena gracilis using ethyl methanesulfonate (EMS) mutagenesis. While the experiments are well-executed, some revisions are necessary before publication:

The importance and advantages of prolonged EMS exposure should be elaborated upon with appropriate references. The novelty of the manuscript cannot rely solely on microalgal mutation; additional context is needed to highlight its significance.

Sol) Thank you for the insightful comment. Long-term treatment with EMS is a critical strategy for increasing the frequency of mutation induction, allowing for the acquisition of a diverse range of mutants. Serrat X, Esteban R, Guibourt N, Moysset L, Nogués S, Lalanne E. EMS mutagenesis in mature seed-derived rice calli as a new method for rapidly obtaining TILLING mutant populations. Plant methods. 2014,10,1-14. Although prolonged EMS exposure enhances the probability of inducing specific mutations, excessive exposure may compromise cell survival. Therefore, it is crucial to balance concentration and duration to maximize mutation efficiency depending on the species.

Figures 1 and 2 suggest that microscopy observations are sufficient to assess mutation conditions. However, the explanation provided in the text is inadequate and requires further elaboration to support this conclusion.

Sol) To provide a more detailed explanation of Figures 1 and 2, the related sentences were revised, and additional supporting sentences were added.

About Figure 1 and Table 1

“In this study, Euglena showed a survival rate of 3.33% when treated with 0.012 M EMS for 72 hours.”

“Interestingly, the reduction in Euglena motility (Table 1) and survival rate (Fig. 1) shows a similar pattern, proportional to the EMS treatment concentration and duration.”

“After treatment with 0.016 M EMS for 48 hours, Euglena motility was not observed under the microscope (Table 1), and no Euglena colonies were detected (Fig. 1)”

“When Euglena exhibited ++ (about 40%) motility, its survival rate ranged from 15.60% to 18.15%.”

The revised parts are highlighted in red color in the text of this manuscript (Line 216 - 222 on Page 5).

About Figure 2

We identified mistake in the description related to Figure 2.

“After 3 days the colony of Mutant 333 was much larger than that of WT, and after 7 days Mutant 333 had a colony diameter of 7 cm, nearly twice that of the WT (3.8 cm).”

The above sentence has been revised as follows:

After 3 days the colony of Mutant 333 was much larger than that of WT, and after 7 days Mutant 333 had a colony diameter of 7.0 cm, nearly 1.3 times larger than that of the WT (5.2 cm).

The revised parts are highlighted in red color in the text of this manuscript (Line 241 - 243 on Page 6).

Statistical analysis results should be included in Figure 7 to enhance the reliability and interpretation of the data.

Sol) Thank you for the suggestion. We added statistical indications in Figure 7 and legend, using SAS ver. 9.4 (SAS Institute Inc., Cary, NC, USA) to enhance the reliability and interpretation of the data.

“Data were analyzed using one-way ANOVA followed by Duncan’s test (p<0.05).”

The revised parts are highlighted in red color in Figure 7 (Line 323 - 324 on Page 10).

The significance of the GC data and its implications for the commercialization potential of the mutants should be discussed in greater detail.

Sol) Particularly, some microalgae can emit strong ‘fishy’ odors which limits their commercial utilization. Many volatile compounds such as alcohols, aldehydes, sulfur compounds, and hydrocarbons, and this was likely responsible for its strong and complex ‘fishy’ aroma. Mutant 335 had no ‘mugwort’ scent (1-octen-3-ol, 2-methyl-1-heptene, styrene), and lacked the ‘barley tea’ and ‘green tea’ scent (hexanal, hexyl formate, 4-ethylbenzoic acid). However, compounds associated with an ‘herbal’ scent (2-methyl-1-undecanol) and a ‘green’ scent (2,4,6-Trimethyldecane) had higher concentrations in Mutant 335. Our analysis of volatile substances showed that Mutant 335 had lower levels of aldehydes and sulfur compounds than the WT, so that Mutant 335 may make it suitable for various food products, such as tea, salad dressings, and herb-based foods.

We have added the above content in the text of the manuscript.

The revised parts are highlighted in red color in the text of this manuscript (Line 373 - 379 on Page 11).

Reviewer 2 Report

Comments and Suggestions for Authors

This study developed effective conditions for generating Euglena mutants by applying a low concentration of EMS over an extended period and using serial treatment to enhance survival rates. The study demonstrated that the cell count and activity observed on agar plates were consistent with results in liquid cultures, allowing for the development of a screening method to select superior strains without the need for liquid cultivation. Among the generated mutants, the highly motility Euglena strain exhibited reduced paramylon productivity. Additionally, the pigment-deficient strain exhibited high paramylon content and reduced volatile compounds, suggesting its potential for commercial applications in the food industry. These findings provide valuable insights for related research and seem highly suitable for publication in this journal. However, a few revisions are recommended.

Major revision

1.      Include statistical data in Figure 1: add statistical values to the survival rate data in Figure 1 to provide more robust and quantitative insights.

2.      Consistent terminology for "motility": standardize the term "motility" throughout the manuscript, replacing "motile" where appropriate (line 92).

3.      Indicate colony diameter in figures related to cell motility: add measurements of colony diameters in figures showing cell motility to provide clearer visual evidence.

4.      Define "glucose-free liquid medium" in Figure 6: clarify what "glucose-free liquid medium" refers to in the methods or figure description, as this term is not explicitly defined.

5.      Add standard deviation to cell concentrations (line 252): include the standard deviation for cell concentrations to enhance data reliability and reproducibility.

6.      Clarify duration for cell density calculations: verify whether the duration for cell density calculations in the methodology should be "6 days" or "7 days," as the current discrepancy creates confusion.

7.      Specify culture media for paramylon measurements: in the paramylon measurement methodology, explicitly state which media were used for cultivating the samples analyzed.

8.      Discuss increased paramylon content in pigment deficient mutants: provide a scientific discussion on why the lack of pigments in Mutant 335 correlates with an increase in paramylon content.

9.      Correct mugwort scent description for Mutant 335 (line 352): correct the description of "mugwort scent" as it appears that Mutant 335 lacks this scent, contrary to what is currently stated.

10.  Add discussion why Mutant 333 was not analyzed for volatile compounds: provide reasoning for why volatile compound analysis was not conducted for Mutant 333, as this data could contribute to a more comprehensive characterization.

11.  Remove cosmetic-related content: the inclusion of cosmetic applications appears subjective and speculative. It is recommended to remove this content for a more objective focus.

12.  Discuss the reduced volatile compounds in Mutant 335: possible reasons for the lower levels of volatile compounds in the pigment-deficient Mutant 335.

Author Response

Reviewer 2

This study developed effective conditions for generating Euglena mutants by applying a low concentration of EMS over an extended period and using serial treatment to enhance survival rates. The study demonstrated that the cell count and activity observed on agar plates were consistent with results in liquid cultures, allowing for the development of a screening method to select superior strains without the need for liquid cultivation. Among the generated mutants, the highly motility Euglena strain exhibited reduced paramylon productivity. Additionally, the pigment-deficient strain exhibited high paramylon content and reduced volatile compounds, suggesting its potential for commercial applications in the food industry. These findings provide valuable insights for related research and seem highly suitable for publication in this journal. However, a few revisions are recommended.

Major revision

  1. Include statistical data in Figure 1: add statistical values to the survival rate data in Figure 1 to provide more robust and quantitative insights.

Sol) We added the indications of statistic in Figures 1 by SAS ver. 9.4 (SAS Institute Inc., Cary, NC, USA), while statistical indications were added in Fig. 1. Bars (mean ± SEM) with different letters are significantly different (p < 0.05).

We added the corresponding sentence in the legend of Fig.1

Data were analyzed using one-way ANOVA followed by Duncan’s test (p <0.05).

Fig. 1 was replaced with a new Fig. 1 that includes statistical annotations." (Line 225 - 229 on Page 6).

  1. Consistent terminology for "motility": standardize the term "motility" throughout the manuscript, replacing "motile" where appropriate (line 92).

Sol) Thank you for the suggestion. The term 'motile' was revised to 'motility' throughout the manuscript to ensure consistency in terminology. (see Line 99)

The revised parts are highlighted in red color in the text of this manuscript (Line 99 on Page 3).

  1. Indicate colony diameter in figures related to cell motility: add measurements of colony diameters in figures showing cell motility to provide clearer visual evidence.

Sol) Indication of diameter of the colony was added to Figure 2.

Fig. 2 was replaced with a new Fig. 2 (on Page 7).

  1. Define "glucose-free liquid medium" in Figure 6: clarify what "glucose-free liquid medium" refers to in the methods or figure description, as this term is not explicitly defined.

Sol) We detailed the composition of the “glucose-free medium” in section 2.3 of Methods and Materials.

The revised parts are highlighted in red color in the text of this manuscript (Line 108 on Page xx).

  1. Add standard deviation to cell concentrations (line 252): include the standard deviation for cell concentrations to enhance data reliability and reproducibility.

Sol) The experiments were performed in triplicate. We added the statistical indication (standard deviation) in the corresponding sentence of the manuscript: see following sentence.

The experiments were performed in triplicate.

The WT cells had a concentration of 1.76 × 10⁵ ± 47 cells mL-1 and the Mutant 333 cells had a concentration of 0.74 × 10⁵ ± 205 cells mL-1.

The revised parts are highlighted in red color in the text of this manuscript (Line 249 - 250 on Page 7).

  1. Clarify duration for cell density calculations: verify whether the duration for cell density calculations in the methodology should be "6 days" or "7 days," as the current discrepancy creates confusion.

Sol) Thank you for pointing out the discrepancy. We revised “6 days” to “7 days”.

The revised parts are highlighted in red color in the text of this manuscript (Line 101 on Page 3).

  1. Specify culture media for paramylon measurements: in the paramylon measurement methodology, explicitly state which media were used for cultivating the samples analyzed.

Sol) Thank for your comment. Euglena cells grown in modified YM medium was used for the measurement of paramylon contents. We added the corresponding sentence “Euglena cells was grown in modified YM medium.” in section 2.5 and 2.6 of Methods and Materials.

The revised parts are highlighted in red color in the text of this manuscript (Line 124 on Page 3 and Line 136 on Page 4).

  1. Discuss increased paramylon content in pigment deficient mutants: provide a scientific discussion on why the lack of pigments in Mutant 335 correlates with an increase in paramylon content.

Sol) In the present study, the pigment-deficient Mutant 335 accumulated approximately 1.72-times more paramylon than WT. The deficiency in pigments such as chlorophyll reduces the efficiency of photosynthesis, potentially redirecting carbon flux towards storage compounds like paramylon as a compensatory mechanism [18]. Pigment-deficient mutant exhibits increased carbon storage as paramylon, likely as an adaptive response to reduced light energy capture. Under pigment-deficient conditions, cells might prioritize the accumulation of paramylon to cope with metabolic stress and maintain cellular homeostasis [29]. In other words, the reduced pigment content may minimize the energy demand re-quired for light harvesting, resulting in the promotion of the synthesis and accumulation of storage carbohydrates such as paramylon.

We included this discussion in the manuscript.

The revised parts are highlighted in red color in the text of this manuscript (Line 325 - 334 on Page 10 -11).

  1. Correct mugwort scent description for Mutant 335 (line 352): correct the description of "mugwort scent" as it appears that Mutant 335 lacks this scent, contrary to what is currently stated.

Sol) Thank you for your comment, and we have revised this section as follows: “Notably, Mutant 335 had no ‘mugwort’ scent (1-octen-3-ol, 2-methyl-1-heptene, styrene), and lacked the ‘barley tea’ and ‘green tea’ scent (hexanal, hexyl formate, 4-ethylbenzoic acid) that was present in WT.”

The revised parts are highlighted in red color in the text of this manuscript (Line 373 - 375 on Page 11).

  1. Add discussion why Mutant 333 was not analyzed for volatile compounds: provide reasoning for why volatile compound analysis was not conducted for Mutant 333, as this data could contribute to a more comprehensive characterization.

Sol) Thank you for the comment. Mutant 333 was not included in the volatile compound analysis because, upon sensory evaluation, its odor did not show any noticeable difference compared to the WT strain.

  1. Remove cosmetic-related content: the inclusion of cosmetic applications appears subjective and speculative. It is recommended to remove this content for a more objective focus.

Sol) Thank you for the suggestion. I agree that the fragrance aspect in cosmetic-related applications can be highly subjective. As such, this content was removed from the manuscript to maintain a more objective focus.

The revised parts are highlighted in red color in the text of this manuscript (Line 377 - 379 on Page 11).

  1. Discuss the reduced volatile compounds in Mutant 335: possible reasons for the lower levels of volatile compounds in the pigment-deficient Mutant 335.

Sol) We appreciate your comments. We also thought it was important to find a direct connection between volatile compounds and the photosynthetic mechanism. In pigment-deficient Mutant 335, photosynthesis is not possible, which is why we believe it is important to investigate volatile compounds associated with photosynthesis. Although we faced difficulties in finding literature directly linking volatile compounds with photosynthetic mechanisms, we found that more volatile compounds are produced in oils like linoleic acid when chlorophyll is present. In principle, chlorophyll can generate singlet oxygen from triplet oxygen upon the exposure of fluorescent light, which plays important roles in the formation of 2-pentylfuran, trans-2-heptenal, and 1-octen-3-ol in linoleic acid. 1-Octen-3-ol is one of the volatile compounds found in the WT in our Euglena-related research but is not detected in the pigment-deficient Mutant 335. In Mutant 335, chlorophyll-photosensitized reactions do not occur, which may explain the reduced concentration and number of volatile compounds observed compared to the WT. However, the possibility of production of volatile compound caused by other genetic changes resulting from EMS treatment cannot be ignored, it is also limited to explain the extensive changes in the volatile compound profile solely by genetic variations.

We included this discussion in the manuscript.

The revised parts are highlighted in red color in the text of this manuscript (Line 356 - 370 on Page 11).

Reviewer 3 Report

Comments and Suggestions for Authors

Dear Authors,

I believe the authors should conduct a thorough review of many critical aspects of the paper. Results and discussion mainly.

The biggest weakness:

In this paper  the authors do not demonstrate that the observed phenotype in the mutants is related to the mutations they claim. For example, they have a mutant with reduced mobility, yes, but due to the way it was generated (repeated mutation cycles with EMS), this mutant could have hundreds of other mutated genes unrelated to mobility that might actually be responsible for the phenotype of reduced paramylon content. The same issue applies to the other mutant: it lacks chlorophyll, yes, but it could also have hundreds of other mutated genes, one of which might be responsible for the phenotype of low paramylon content. In other words, in my opinion, they do not demonstrate the relationship between the phenotypes and the mutations they are supposed to have.

Majors:

- In the introduction, it is not explained what advantages this microalga has over other model algae.

-In the introduction, I miss the explanation of the mechanism of action of EMS as a mutagen; please include it.

-Important, Figure 1 lacks both error bars and statistical error; both should be included

-L231: “The WT cells had a concentration of 1.76 × 10⁵ cells mL-1 and the Mutant 333 cells had a concentration of 0.74 × 10⁵ cells mL-1“ Have you performed this experiment only once? How many times have you conducted it? It is essential that all experiments are carried out at least three times and include their corresponding statistical errors, pleaase.

-L235: “to multiple mutations” This is an important point. By applying three repetitive treatments with EMS, you are generating a large number of distinct mutations, maybe hundreds. These isolated mutants are likely to have numerous different metabolic pathways altered. Please discuss this in greater depth, please.

-L260: “We attribute these results  to the greater motility of Mutant 333 under phototrophic conditions without mixing. This  enhanced motility likely led to increased overall absorbance of light by chlorophyll at high  cell densities (OD680nm > 0.2), resulting in increased photosynthesis” Is there bibliographic support for this hypothesis? If so, please indicate it.

-L270: “two methods”  Why two different methods? What differences do they have? Please explain.

-L276: “likely a consequence of its enhanced motility” Is there bibliographic support for this hypothesis? If so, please indicate it

-L278: “a Euglena mutant generated by Fe ion irradiation” But does it have reduced mobility?

- Which of the volatile compounds in Table 2 are related to photosynthesis? This is crucial to determine if they are connected to the photosynthesis defect in this mutant. The authors should identify them, mark them in Table 2, and discuss this, please.

-L280: “As reduced motility led to increased paramylon accumulation, an increase in motility could similarly result in decreased paramylon content” Is there any hypothesis with bibliographic support to explain this?

-L290: “WZSL strain accumulated” Where does paramylon accumulate? What relationship could it have with photosynthesis?

- The authors should have performed genetic crosses with parental strains of the opposite sex, studied the segregation, and determined if the segregants and mutants also exhibit the observed phenotype, in order to know if the phenotypes are related. By calculating the segregation ratio, they could also have estimated how many genes are involved.

-In my opinion, the discussion is insufficient in the sense that the biotechnological improvement presented by these two strains is not well understood. It should have been discussed in relation to the literature, particularly with other model algae strains such as Chlamydomonas, where their potential in bioremediation and bioproduction has been recently studied,  please discuss this.

Author Response

Reviewer 3

In this paper the authors do not demonstrate that the observed phenotype in the mutants is related to the mutations they claim. For example, they have a mutant with reduced mobility, yes, but due to the way it was generated (repeated mutation cycles with EMS), this mutant could have hundreds of other mutated genes unrelated to mobility that might actually be responsible for the phenotype of reduced paramylon content. The same issue applies to the other mutant: it lacks chlorophyll, yes, but it could also have hundreds of other mutated genes, one of which might be responsible for the phenotype of low paramylon content. In other words, in my opinion, they do not demonstrate the relationship between the phenotypes and the mutations they are supposed to have.

Majors:

- In the introduction, it is not explained what advantages this microalga has over other model algae.

Sol) Euglena gracilis is a green microalga known as a potential candidate for production of commercial products that can be used in the field of food, fiber, feed, fertilizer, and fuel [1]. As an example in the food industry, these cells can produce many high-value metabolic products, including vitamins, amino acids, pigments, unsaturated fatty acids, and carbo-hydrates [1-3]. Especially, Euglena gracilis can produce unique substances, such as para-mylon and wax esters, that cannot be biosynthesized by other microorganism. Paramylon is a linear β-1,3-glucan that has potential as a functional product due to its im-mune-enhancing, cholesterol-reducing, and antioxidant activity [4-8]. Euglena also con-verts paramylon into wax esters in the cytoplasm under anaerobic conditions, and these wax esters compose entirely of saturated carbon chains, myristyl myristate (C14:0-C14:0), which is an ideal raw material for a drop-in bio jet fuel [9, 10]. In addition, Euglena gracilis possesses the ability to survive in extreme conditions, such as low pH environments, which facilitates its dominance in cultivation systems [11].

[Reference]

[1] He J, Liu C, Du M, Zhou X, Hu Z, Lei A, et al. Metabolic responses of a model green microalga Euglena gracilis to different environmental stresses. Frontiers in Bioengineering and Biotechnology. 2021,9,662655.

[2] Kitaoka S, Hosotani K. Studies on culture conditions for the determination of the nutritive value of Euglena gracilis protein and the general and amino acid compositions of the cells. 1977.

[3] Baker ER, McLaughlin JJ, Hutner SH, DeAngelis B, Feingold S, Frank O, et al. Water-soluble vitamins in cells and spent culture supernatants of Poteriochromonas stipitata, Euglena gracilis, and Tetrahymena thermophila. Archives of Microbiology. 1981,129,310-3.

[4] Kim K, Kang J, Seo H, Kim S, Kim DY, Park Y, et al. A novel screening strategy utilizing aniline blue and calcofluor white to develop paramylon-rich mutants of Euglena gracilis. Algal Research. 2024,78,103408.

[5] Carballo C, Chronopoulou EG, Letsiou S, Maya C, Labrou NE, Infante C, et al. Antioxidant capacity and immunomodulatory effects of a chrysolaminarin-enriched extract in Senegalese sole. Fish & shellfish immunology. 2018,82,1-8.

[6] Del Cornò M, Gessani S, Conti L. Shaping the innate immune response by dietary glucans: any role in the control of cancer? Cancers. 2020,12,155.

[7] Stier H, Ebbeskotte V, Gruenwald J. Immune-modulatory effects of dietary Yeast Beta-1, 3/1, 6-D-glucan. Nutrition journal. 2014,13,38.

[8] Harada R, Nomura T, Yamada K, Mochida K, Suzuki K. Genetic engineering strategies for Euglena gracilis and its industrial contribution to sustainable development goals: A review. frontiers in Bioengineering and Biotechnology. 2020,8,556462.

[9] Teerawanichpan P, Qiu X. Fatty acyl-CoA reductase and wax synthase from Euglena gracilis in the biosynthesis of medium-chain wax esters. Lipids. 2010,45,263-73.

[10] Bakku RK, Yamamoto Y, Inaba Y, Hiranuma T, Gianino E, Amarianto L, et al. New insights into raceway cultivation of Euglena gracilis under long-term semi-continuous nitrogen starvation. Scientific Reports. 2023,13,7123.

[11] Gissibl A, Sun A, Care A, Nevalainen H, Sunna A. Bioproducts from Euglena gracilis: synthesis and applications. Frontiers in bioengineering and biotechnology. 2019,7,108.

We explained the advantages of this microalga over other algae.

The revised parts are highlighted in red color in the text of this manuscript (Line 31 -43 on Page 1 - 2).

- In the introduction, I miss the explanation of the mechanism of action of EMS as a mutagen; please include it.

Sol) The chemical mutagen EMS, an alkylating agent, is specific to guanine. In principle, EMS can easily affects DNA, causing mutations by modifying the nucleotide bases. It reacts with purine bases, particularly adenine (A) and guanine (G), leading to incorrect base pairing during DNA replication.

We added the mechanism of EMS in the text of this manuscript in way to structure it to fit the overall sentence content: see following sentence.

“The chemical mutagen EMS, an alkylating agent, which reacts with purine bases, particularly adenine (A) and guanine (G), leading to incorrect base pairing during DNA replication [17]. In principle, EMS can easily affects DNA in Euglena. However, very few previous studies have used EMS mutagenesis to develop useful strains of Euglena.”

[Reference]

[17] Trovão M, Schüler LM, Machado A, Bombo G, Navalho S, Barros A, et al. Random mutagenesis as a promising tool for microalgal strain improvement towards industrial production. Marine drugs. 2022,20,440.

The revised parts are highlighted in red color in the text of this manuscript (Line 56 - 59 on Page 2).

Important, Figure 1 lacks both error bars and statistical error; both should be included

Sol) We added the indications of statistic in Figures 1 by SAS ver. 9.4 (SAS Institute Inc., Cary, NC, USA), while statistical indications were added in Fig. 1. Bars (mean ± SEM) with different letters are significantly different (p < 0.05).

We added the corresponding sentence in the legend of Fig.1

Data were analyzed using one-way ANOVA followed by Duncan’s test (p <0.05).

Fig. 1 was replaced with a new Fig. 1 that includes statistical annotations." (on Page 6).

-L231: “The WT cells had a concentration of 1.76 × 10⁵ cells mL-1 and the Mutant 333 cells had a concentration of 0.74 × 10⁵ cells mL-1“ Have you performed this experiment only once? How many times have you conducted it? It is essential that all experiments are carried out at least three times and include their corresponding statistical errors, please.

Sol) The experiments were performed in triplicate. We added the statistical indication (standard deviation) in the corresponding sentence of the manuscript: see following sentence.

The experiments were performed in triplicate.

The WT cells had a concentration of 1.76 × 10⁵ ± 47 cells mL-1 and the Mutant 333 cells had a concentration of 0.74 × 10⁵ ± 205 cells mL-1.

The revised parts are highlighted in red color in the text of this manuscript (Line 249 - 250 on Page 7).

-L235: “to multiple mutations” This is an important point. By applying three repetitive treatments with EMS, you are generating a large number of distinct mutations, maybe hundreds. These isolated mutants are likely to have numerous different metabolic pathways altered. Please discuss this in greater depth, please.

Sol) We appreciate your comment, and totally agree with you. EMS mutagenesis faces limitations due to its random mutation patterns, making it difficult to target specific genes. Furthermore, as it primarily induces point mutations, identifying and analyzing these mutations requires significant time and advanced techniques. Mutant 333 shows the increased motility as phenotype, and Mutant 335 shows less chlorophyll. Iit is difficult to explain the changes in paramylon content in species with increased mobility (Mutant 333) or lack of pigmentation (Mutant 335) through the phenotype they exhibit, as it may be due to other genetic mutations caused by EMS treatment. So, although we cannot completely rule out the possibility that other genetic mutations may be responsible for the changes in paramylon content, we tried to revise the paper by providing a comparative explanation with literature studies showing similar patterns of metabolism in same or completely opposite phenotypes in transformed Euglena strain and other microalgae including Chlamydomonas reinhardtii which you recommended for reference [32]. We have tried to avoid misinterpretation by other readers by suggesting that the results of this change in paramylon content may be due to other genetic defects, along with the inference of results through comparative studies based on the literature.

[Reference]

[32] Hamilton BS, Nakamura K, Roncari DA. Accumulation of starch in Chlamydomonas reinhardtii flagellar mutants. Biochemistry and Cell Biology. 1992,70,255-8.

This study focused on the development of effective conditions for generating Euglena mutants by EMS treatment, in way applying a low concentration of EMS over an extended period and using serial treatment to enhance survival rates. The study demonstrated that the cell count and activity observed on agar plates were consistent with results in liquid cultures, allowing for the development of a screening method to select superior strains without the need for liquid cultivation. We hope that you consider the limitations of EMS research so that many more researchers can easily develop improved Euglena species through the newly developed EMS method. For the questions below, we did our best to explain them through comparative studies with similar results.

-L260: “We attribute these results to the greater motility of Mutant 333 under phototrophic conditions without mixing. This enhanced motility likely led to increased overall absorbance of light by chlorophyll at high cell densities (OD680nm > 0.2), resulting in increased photosynthesis” Is there bibliographic support for this hypothesis? If so, please indicate it.

Sol) According to the Beer-Lambert Law, when the absorbance is greater than 0.2, the initial light intensity decreases by 50% in the center [27]. Therefore, in various experiments opti-mizing cultivation conditions through continuous culture at a constant concentration, the cell density value is adjusted to not exceed an OD680nm value of 0.2 [28, 29]. In the cultiva-tion of Euglena, without physical agitation and with light provided from above, the mixing of internal cells during cultivation relies solely on the motility of Euglena. Based on the results shown in Figure 6, when the OD680nm is below about 0.2, the shading caused by cells within the cultivation space is minimal. Therefore, regardless of Euglena's motility, all Euglena cells in the space can evenly utilize the available light without the issues. As a re-sult, the wild-type (WT), which has a faster division rate than Mutant 333, exhibits a steep growth rate. When the OD680nm value exceeds 0.3, the increase in dark zones caused by self-shading results in light utilization being dependent on the motility of Euglena. Con-sequently, although the WT has a higher division rate, the reduced light utilization allows Mutant 333 to achieve a higher growth rate. As an example of a study demonstrating how the efficiency of Euglena growth changes depending on light utilization, the growth rate and productivity of Euglena gracilis improved as the proportion of light exposure increased, that the highest productivity was observed under the 16:8 light/dark cycle condition com-pared to conditions with shorter light cycles [30].

[Reference]

[27] Devi MP, Subhash GV, Mohan SV. Heterotrophic cultivation of mixed microalgae for lipid accumulation and wastewater treatment during sequential growth and starvation phases: effect of nutrient supplementation. Renewable energy. 2012,43,276-83.

[28] Kwon J-H, Rögner M, Rexroth S. Direct approach for bioprocess optimization in a continuous flat-bed photobioreactor system. Journal of biotechnology. 2012,162,156-62.

[29] Kwon J-H, Bernat G, Wagner H, Roegner M, Rexroth S. Reduced light-harvesting antenna: consequences on cyanobacterial metabolism and photosynthetic productivity. Algal Research. 2013,2,188-95.

[30] Kim S, Lim D, Lee D, Yu J, Lee T. Valorization of corn steep liquor for efficient paramylon production using Euglena gracilis: The impact of precultivation and light-dark cycle. Algal Research. 2022,61,102587.

The revised parts are highlighted in red color in the text of this manuscript (Line 278 - 296 on Page 9).

-L270: “two methods” Why two different methods? What differences do they have? Please explain.

Sol) The principle of the gravimetric method is based on the precise measurement of mass to quantify a specific substance or component in a sample. In this method, the target substance is isolated from the sample, either by precipitation, filtration, or drying. However, errors may occur due to incomplete precipitation, loss of material during washing, or contamination. Aniline blue method offers high specificity for β-1,3-glucan. In most studies on Euglena, the gravimetric method is commonly used to measure paramylon content. Therefore, to infer comparable values with other studies, this study derived results using two different methods.

-L276: “likely a consequence of its enhanced motility” Is there bibliographic support for this hypothesis? If so, please indicate it

Sol) The lower paramylon content in Mutant 333 would be likely a consequence of its enhanced motility. A previous study described a Euglena mutant with flagella defects generated by Fe-ion irradiation that accumulated approximately 1.6-times more paramylon than the it’s WT. This study explains the increase in paramylon accumulation as being caused by the energy conservation by not swimming [31]. As another study, in Chlamydomonas reinhardtii flagellar-deficient mutants, starch accumulation increased due to the absence of energy-intensive processes like flagellar regeneration [32]. Through reverse result caused by completely opposite factor, the greater energy consumption used for motility may have reduced the accumulation of paramylon, a carbohydrate storage compound. However, the possibility of changes in paramylon content caused by other genetic defects resulting from EMS treatment cannot be ignored, and further research is needed.

[Reference]

[31] Muramatsu S, Atsuji K, Yamada K, Ozasa K, Suzuki H, Takeuchi T, et al. Isolation and characterization of a motility-defective mutant of Euglena gracilis. PeerJ. 2020,8,e10002.

[32] Hamilton BS, Nakamura K, Roncari DA. Accumulation of starch in Chlamydomonas reinhardtii flagellar mutants. Biochemistry and Cell Biology. 1992,70,255-8.

The revised parts are highlighted in red color in the text of this manuscript (Line 309 - 320 on Page 10).

 - L278: “a Euglena mutant generated by Fe-ion irradiation” But does it have reduced mobility?

Sol) The mutant Euglena gracilis strain generated through Fe ion irradiation exhibited significantly reduced motility compared to the wild-type strain, primarily due to defects in the flagella.

The revised parts are highlighted in red color in the text of this manuscript (Line 310 - 314 on Page 10).

- Which of the volatile compounds in Table 2 are related to photosynthesis? This is crucial to determine if they are connected to the photosynthesis defect in this mutant. The authors should identify them, mark them in Table 2, and discuss this, please.

Sol) We appreciate your comments. We also thought it was important to find a direct connection between volatile compounds and the photosynthetic mechanism. In pigment-deficient Mutant 335, photosynthesis is not possible, which is why we believe it is important to investigate volatile compounds associated with photosynthesis. Although we faced difficulties in finding literature directly linking volatile compounds with photosynthetic mechanisms, we found that more volatile compounds are produced in oils like linoleic acid when chlorophyll is present. In principle, chlorophyll can generate singlet oxygen from triplet oxygen upon the exposure of fluorescent light, which plays important roles in the formation of 2-pentylfuran, trans-2-heptenal, and 1-octen-3-ol in linoleic acid [37]. 1-Octen-3-ol is one of the volatile compounds found in the WT in our Euglena-related research but is not detected in the pigment-deficient Mutant 335. In Mutant 335, chlorophyll-photosensitized reactions do not occur, which may explain the reduced concentration and number of volatile compounds observed compared to the WT. However, the possibility of production of volatile compound caused by other genetic changes resulting from EMS treatment cannot be ignored, it is also limited to explain the extensive changes in the volatile compound profile solely by genetic variations.

[Reference]

[37] Lee J, Min DB. Analysis of volatile compounds from chlorophyll photosensitized linoleic acid by headspace solid-phase microextraction (HS-SPME). Food Science and Biotechnology. 2010,19,611-6.

The revised parts are highlighted in red color in the text of this manuscript (Line 356 - 368 on Page 11).

-L280: “As reduced motility led to increased paramylon accumulation, an increase in motility could similarly result in decreased paramylon content” Is there any hypothesis with bibliographic support to explain this?

Sol) The lower paramylon content in Mutant 333 would be likely a consequence of its enhanced motility. A previous study described a Euglena mutant with flagella defects generated by Fe-ion irradiation that accumulated approximately 1.6-times more paramylon than the it’s WT. This study explains the increase in paramylon accumulation as being caused by the energy conservation by not swimming [31]. As another study, in Chlamydomonas reinhardtii flagellar-deficient mutants, starch accumulation increased due to the absence of energy-intensive processes like flagellar regeneration [32]. Through reverse result caused by completely opposite factor, the greater energy consumption used for motility may have reduced the accumulation of paramylon, a carbohydrate storage compound. However, the possibility of changes in paramylon content caused by other genetic defects resulting from EMS treatment cannot be ignored, and further research is needed.

[Reference]

[31] Muramatsu S, Atsuji K, Yamada K, Ozasa K, Suzuki H, Takeuchi T, et al. Isolation and characterization of a motility-defective mutant of Euglena gracilis. PeerJ. 2020,8,e10002.

[32] Hamilton BS, Nakamura K, Roncari DA. Accumulation of starch in Chlamydomonas reinhardtii flagellar mutants. Biochemistry and Cell Biology. 1992,70,255-8.

The revised parts are highlighted in red color in the text of this manuscript (Line 309 - 320 on Page 10).

-L290: “WZSL strain accumulated” Where does paramylon accumulate? What relationship could it have with photosynthesis?

Sol) WZSL mutant is a non-photosynthetic strain lacking chloroplasts, and it accumulates paramylon granules in the cytoplasm. This mutant can accumulate paramylon up to 90% of its dry weight when using glucose as a carbon source under dark conditions. In the present study, the pigment-deficient Mutant 335 accumulated approximately 1.72-times more paramylon than WT. The deficiency in pigments such as chlorophyll re-duces the efficiency of photosynthesis, potentially redirecting carbon flux towards storage compounds like paramylon as a compensatory mechanism [18]. Pigment-deficient mutant exhibits increased carbon storage as paramylon, likely as an adaptive response to reduced light energy capture. Under pigment-deficient conditions, cells might prioritize the accumulation of paramylon to cope with metabolic stress and maintain cellular homeostasis [29]. In other words, the reduced pigment content may minimize the energy demand required for light harvesting, resulting in the promotion of the synthesis and accumulation of storage carbohydrates such as paramylon.

[Reference]

[18] Shin W-S, Lee B, Jeong B-r, Chang YK, Kwon J-H. Truncated light-harvesting chlorophyll antenna size in Chlorella vulgaris improves biomass productivity. Journal of Applied Phycology. 2016,28,3193-202.

[29] Kwon J-H, Bernat G, Wagner H, Roegner M, Rexroth S. Reduced light-harvesting antenna: consequences on cyanobacterial metabolism and photosynthetic productivity. Algal Research. 2013,2,188-95.

The revised parts are highlighted in red color in the text of this manuscript (Line 325 - 334 on Page 10 - 11).

- The authors should have performed genetic crosses with parental strains of the opposite sex, studied the segregation, and determined if the segregants and mutants also exhibit the observed phenotype, in order to know if the phenotypes are related. By calculating the segregation ratio, they could also have estimated how many genes are involved.

Sol) Thank you for your advice. Due to the limitations of utilizing mutations induced by EMS in Euglena research, implementing various genetic approaches such as genetic crosses, as you suggested, has been challenging in our current research environment.

-In my opinion, the discussion is insufficient in the sense that the biotechnological improvement presented by these two strains is not well understood. It should have been discussed in relation to the literature, particularly with other model algae strains such as Chlamydomonas, where their potential in bioremediation and bioproduction has been recently studied, please discuss this.

Sol) We found study reporting an increase in the content of starch as storage materials like paramylon, in Chlamydomonas strains with damaged flagella, In Chlamydomonas reinhardtii flagellar-deficient mutants, starch accumulation increases due to the absence of energy-intensive processes like flagellar regeneration [32]. This reduction in energy consumption during carbon metabolism leads to excess production of starch like paramylon.

[Reference]

[32] Hamilton BS, Nakamura K, Roncari DA. Accumulation of starch in Chlamydomonas reinhardtii flagellar mutants. Biochemistry and Cell Biology. 1992,70,255-8.We added the corresponding sentence in the text of manuscript

The revised parts are highlighted in red color in the text of this manuscript (Line 314 - 316x on Page 10).

Reviewer 4 Report

Comments and Suggestions for Authors

The study utilized chemical method to obtain two mutated strains of E. gracilis which present specific characteristics. The study is well designed and  key results are logically analyzed and discussed. And will attract interests from potential readers.

Minor suggestion:

(1) And the end of abstract, add one short sentences to discuss tentative future application of the mutated strains. 

(2) It is very rare to see such a long table (table 2) in context. The data can be removed to supplementary file and only left several key data in context.

(3) Although molecular data of the mutation of the two strains is lacked in the study. Could it be further discussed in results and discussion section??? 

Author Response

Reviewer 4

The study utilized chemical method to obtain two mutated strains of E. gracilis which present specific characteristics. The study is well designed and key results are logically analyzed and discussed. And will attract interests from potential readers.

Minor suggestion:

(1) And the end of abstract, add one short sentence to discuss tentative future application of the mutated strains. 

Sol) Thank you for the suggestion. We added a corresponding sentence at the end of the abstract

“The Mutant 335 strain is suitable for the production of functional food products and renewable jet fuel.”

The revised parts are highlighted in red color in the text of this manuscript (Line 25 – 26 on Page 1).

(2) It is very rare to see such a long table (table 2) in context. The data can be removed to supplementary file and only left several key data in context.

Sol) We moved Table 2 from the main text to Supplementary Table 1, and updated the table label accordingly to match the changes.

(3) Although molecular data of the mutation of the two strains is lacked in the study. Could it be further discussed in results and discussion section??? 

Sol) Thank you for comment. The molecular data about the two mutant strains is not included in this study. Identifying and analyzing these mutations requires significant time and advanced techniques. Therefore, it is quite challenging for us to add the relevant molecular data within the given revision period.

This study focused on the development of effective conditions for generating Euglena mutants by EMS treatment, in way applying a low concentration of EMS over an extended period and using serial treatment to enhance survival rates. The study demonstrated that the cell count and activity observed on agar plates were consistent with results in liquid cultures, allowing for the development of a screening method to select superior strains without the need for liquid cultivation.

If possible, we would appreciate it if you could allow us to add the following sentence in Results and Discussion section to highlight the importance of molecular biological analysis.

“In the future, identification of the responsible gene in the mutants can provide a clearer understanding of the relationship between the associated phenotypes and changes in paramylon and volatile compound content.”

The revised parts are highlighted in red color in the text of this manuscript (Line 368 - 370 on Page 11).

Round 2

Reviewer 3 Report

Comments and Suggestions for Authors

Dear Authors,

I believe the authors have adequately addressed all of my comments and suggestions, and I accept the paper in its current version.